# Associations of Cooking Skill with Social Relationships and Social Capital among Older Men and Women in Japan: Results from the JAGES

**DOI:** 10.3390/ijerph20054633

**Published:** 2023-03-06

**Authors:** Yukako Tani, Takeo Fujiwara, Katsunori Kondo

**Affiliations:** 1Department of Global Health Promotion, Tokyo Medical and Dental University (TMDU), 1-5-45, Yushima, Bunkyo-ku, Tokyo 113-8519, Japan; 2Department of Social Preventive Medical Sciences, Center for Preventive Medical Sciences, Chiba University, 1-8-1 Inohana, Chuo-ku, Chiba-shi 260-8672, Japan; 3Department of Gerontological Evaluation, Center for Gerontology and Social Science, National Center for Geriatrics and Gerontology, 7-430 Morikoka-cho, Obu-shi 474-8511, Japan

**Keywords:** cooking skill, social relationship, social capital, older adults

## Abstract

The health benefits of social relationships and social capital are well known. However, little research has examined the determinants of social relationships and social capital. We examined whether cooking skill was associated with social relationships and social capital in older Japanese people. We used 2016 Japan Gerontological Evaluation Study data on a population-based sample of men and women aged ≥ 65 years (*n* = 21,061). Cooking skill was assessed using a scale with good validity. Social relationships were evaluated by assessing neighborhood ties, frequency and number of meetings with friends, and frequent meals with friends. Individual-level social capital was evaluated by assessing civic participation, social cohesion, and reciprocity. Among women, high-level cooking skill was positively associated with all components of social relationships and social capital. Women with high-level cooking skill were 2.27 times (95% CI: 1.77–2.91) more likely to have high levels of neighborhood ties and 1.65 (95% CI: 1.20–2.27) times more likely to eat with friends, compared with those with middle/low-level cooking skill. Cooking skills explained 26.2% of the gender difference in social relationships. Improving cooking skills may be key to boosting social relationships and social capital, which would prevent social isolation.

## 1. Introduction

Globally, there were 901 million people aged 60 years or older in 2015, and this number is projected to rise to 1.4 billion by 2030 [1]. In older age, social networks may decline because of retirement, adult children’s independence, and bereavement after the death of spouses or friends. Socially isolated older people are at increased risk of several detrimental health outcomes including mortality [2], dementia [3], and poor mental health [4]. Therefore, it is important to find modifiable factors that foster social relationships among older adults.

Social relationships are measured in a variety of ways, with three main aspects being used in research on health: social network, social activity, and social support [5]. Social networks and social activity represent structural aspects of social relationships, whereas social support represents functional aspects of social relationships [5]. Social network covers network size (number of members) and density (frequency of contact between members), social activities are represented by social participation and social engagement, and social support refers to a perception of the availability of support from members of the social network [5]. In addition to social relationships, social capital is another important health-promoting concept. Social capital is described as resources that people can receive through their social networks, although there is no universally agreed definition of social capital [6,7].

The health benefits of both Individual- and community-level social capital have been shown in many epidemiological studies [6,7,8,9]. However, little research has examined the determinants of social capital. Recently, gender inequality in social capital has been reported, with women having higher levels of some social capital components, such as reciprocity and bridging, compared with men [10,11]. Compared with men, women tend to invest more in social relationships and building intimate emotional relationships [12,13]. In a study of older adults in Japan and England, women more often met with friends than did men [14,15]. However, the reasons for differences in social relationships between men and women are still unknown. In addition to gender, ethnicity and socioeconomic status (SES) have been reported as possible determinants of social capital [10,11,16], but these factors (i.e., gender, ethnicity, and SES) are difficult or impossible to modify through intervention. To boost social capital, modifiable factors determining social relationships and social capital should be identified.

Activity-related food has been linked with social activity from an evolutionary perspective [17]. Meal preparation ability may contribute to fostering not only family relationships but also social relationships with neighbors and friends. A qualitative study among rural older adults in the United States reported that most older adults gave or received some kind of food, especially cooked foods and garden products, and women were more likely to receive food gifts than men [18]. This food sharing was valued as a way to maintain reciprocity in social relations and to create a feeling of community membership [18]. In Japan, there is a culture of *osusowake*, which refers to the mutual exchange of foodstuffs between neighbors. This culture may contribute to strengthening community networks through supporting cultural activities including local festivals and seasonal events [19]. A systematic report on the benefits of cooking interventions showed that community kitchen programs had a positive influence on socialization [20]. Higher levels of cooking skills have been found to increase the frequency of cooking and confidence in cooking [21,22,23,24,25]. Thus, cooking skill may increase opportunities to build better social relationships with others, such as sharing food with neighbors and attending local cultural activities.

Cooking skills represent a basic living ability that contributes to better diet quality. Several studies have shown the dietary benefits of cooking skills, such as higher consumption of vegetables and fruits and lower consumption of prepared meals, convenience foods, and ultra-processed foods [21,25,26,27]. However, little is known about the importance of cooking skills beyond dietary outcomes. Although one’s mother is the most common source for learning cooking skills, people also learn from partners, cookbooks, television shows, and cooking classes [23,28]. Thus, interventions are possible even in older age. In fact, because retirement allows more time to cook, it is reasonable for older people to newly start to learn cooking skills.

The aim of this study was to examine the associations of cooking skills with social relationships and social capital among older adults. First, to identify social relationships that can be modified through intervention, we examined the association of cooking skills with social relationships with neighbors and friends rather than with relatives. Specifically, the investigated social relationships included neighborhood ties, frequency of meetings with friends, number of meetings with friends, and shared meals with friends. Next, we examined the associations between cooking skills and individual-level social capital, which included civic participation, social cohesion, and reciprocity [29]. Finally, we examined gender differences in social relationships and social capital, as well as the mediating role of cooking skills in the associations of gender with social relationships and social capital.

## 2. Materials and Methods

### 2.1. Study Design and Participants

We used data from the Japan Gerontological Evaluation Study (JAGES), which was carried out in 39 municipalities across Japan in 2016. The study targeted community-dwelling older adults without functional disabilities, defined as not being certified as eligible to receive long-term public care insurance system services [30]. From October 2016 to January 2017, self-report questionnaires were mailed to 279,661 adults aged ≥ 65 years, and 196,438 individuals returned the questionnaire (response rate: 70.2%). The survey was conducted using random sampling in 22 large municipalities and was administered to all eligible residents in 17 small municipalities [25]. One-eighth of the target sample (*n* = 22,219) were randomly selected to receive the survey module inquiring about cooking skills. Of the 21,061 participants who had information on both gender and cooking skills and did not report any limitations in activities of daily living, those who had information on each outcome variable were included in the analysis; thus, the analytic sample differs depending on the outcome: *n* = 20,799 for neighborhood ties, *n* = 20,477 for frequency of meetings with friends, *n* = 20,445 for the number of meetings with friends, *n* = 21,061 for shared meals with friends, *n* = 15,631 for civic participation, *n* = 20,424 for social cohesion, and *n* = 20,224 for reciprocity. Participants were informed that participation in the study was voluntary and that completing and returning the questionnaire indicated their consent to participate in the study.

### 2.2. Social Relationships

Neighborhood ties, frequency of meetings with friends, number of meetings with friends, and frequent shared meals with friends were evaluated to assess social relationships. All components of social relationships were assessed using the self-report questionnaire. For neighborhood ties, participants were asked, “What kind of interactions do you have with people in your neighborhood?” The four response options were (1) mutual consultation, lending and borrowing daily commodities, and cooperation in daily life; (2) standing and chatting frequently; (3) no more than exchanging greetings; and (4) none, not even greetings [29,31]. We classified the participants as having high (response 1), middle (response 2), or low (response 3 or response 4) levels of ties, collapsing the two response categories because only 2.27% of the participants reported having no interactions with people in their neighborhood (response 4). The frequency of meetings with friends was assessed using the following question: “How often do you see your friends?”. The six response options were (1) ≥4 times/week; (2) 2–3 times/week; (3) 1 time/week; (4) 1–3 times/month; (5) a few times/year; (6) never [29]. In this analysis, the scores of 4, 2.5, 1, 0.5, 0.125, and 0 (times/week) were assigned to response categories 1, 2, 3, 4, 5, and 6, respectively, and the resulting variable was treated as continuous. The number of meetings with friends was assessed using the following question: “How many friends/acquaintances have you seen over the past month?”. The five response options were (1) ≥10; (2) 6–9; (3) 3–5; (4) 1–2; (5) 0 [29]. In this analysis, the scores of 10, 7.5, 4, 1.5, and 0 (persons/month) were assigned to responses 1, 2, 3, 4, and 5, respectively, and the resulting variable was treated as continuous. Frequent shared meals with friends were assessed using the following question: “Who do you usually have meals with?”. The possible responses were no one, spouse, children, grandchildren, friends, and other [32]. Multiple responses were possible. We defined participants who selected “friends” as eating with friends.

### 2.3. Social Capital

Individual-level social capital was evaluated by assessing civic participation, social cohesion, and reciprocity using a validated scale to measure community-level social capital [29]. These variables were assessed using the self-report questionnaire, and details of this assessment have been described elsewhere [29]. For civic participation, we calculated the number of groups in which a respondent participated once or more [29]. Social cohesion was assessed using the following questions: “Do you think people living in your area can be trusted in general?” (community trust), “Do you think most people in your community offer assistance to others?” (norm of reciprocity), and “How strong is your residential place attachment?” (community attachment). Responses were rated on a five-point scale ranging from strongly trusted, agree strongly, or strongly attached to not at all. We calculated the number of items on which the participant strongly or moderately agreed [29]. Reciprocity was assessed using the following questions: “Do you have someone who listens to your concerns and complaints?” (received emotional support), “Do you listen to someone’s concerns and complaints?” (provided emotional support), and “Do you have someone who looks after you when you are sick for a few days?” (received instrumental support). The possible responses were no one, spouse, children, sibling/relative/parent/grandchildren, neighbors, friends, and other. Multiple responses were allowed. To explore the type of reciprocity that can be changed through intervention, we calculated the number of items for which the respondent selected neighbors, friends, or other.

### 2.4. Cooking Skills

Cooking skills were assessed using a cooking skills scale designed with consideration of basic Japanese cooking methods and typical meals; details of this assessment have been described elsewhere [25]. This scale had appropriate internal consistency (Cronbach’s α = 0.96) and notable discriminant validity, with women (experienced food preparers) scoring significantly better than men (food preparation novices) [25]. The scale consisted of seven items: (1) overall cooking skills; (2) able to peel fruits and vegetables; (3) able to boil eggs and vegetables; (4) able to grill fish; (5) able to make stir-fried meat and vegetables; (6) able to make miso soup; and (7) able to make stewed dishes. Participants were asked to evaluate their own cooking skills on a six-point scale ranging from unable (=0) to very well (=5). We calculated the mean of these seven items and divided the result into three categories: high (score of >4.0), middle (score of 2.1–4.0), and low (score of ≤2.0) [25]. For women, the middle group and the low group were combined into one category because the low group was quite small (1.2%).

### 2.5. Covariates

Covariates were assessed using the self-report questionnaire (Appendix A). We included education, current annual household income, and marital status as socio-demographic characteristics [25]. For health status, we asked whether the participants were currently under medical treatment for any of the following conditions: cancer, heart disease, stroke, hypertension, diabetes mellitus, and hyperlipidemia. Furthermore, depressive symptoms were assessed using the Geriatric Depression Scale [33]. To account for personality aspects such as curiosity regarding cooking, which may be directly associated with social relationships, as a sensitivity analysis, we controlled for whether the participants talked with young people [34] and the participants’ willingness to take on a leadership role in a community activity. Participants with missing data on covariates were included in the analysis as dummy variables.

### 2.6. Statistical Analysis

The analyses were stratified by gender because different associations between cooking skills and dietary behaviors have been reported for men and women [25]. First, after stratifying the sample by gender, we tested the differences using the chi-square test for categorical variables and the *t*-test or ANOVA for continuous variables. Next, participants were stratified by their level of cooking skills, and differences were tested using the chi-square test for categorical variables and the *t*-test or ANOVA for continuous variables. Second, for neighborhood ties, we used multinomial logistic regression to calculate adjusted relative risk ratios (RRRs) with 95% CIs of high-level and middle-level ties, with low-level ties as the reference category. For the frequency and number of meetings with friends and social capital (civic participation, social cohesion, and reciprocity), we used multivariate linear regression models, adjusting for potential confounders. For frequent shared meals with friends, we used logistic regression to calculate adjusted odds ratios with 95% CIs of eating meals with friends. The models were adjusted for the following potential confounding factors: age, socio-demographic characteristics (education, annual normalized household income, and marital status), and health status (medical treatment of cancer, heart disease, stroke, hypertension, diabetes mellitus, and hyperlipidemia, as well as depressive symptoms).

Additionally, we conducted structural equation modeling (SEM) analysis to explore the mediating role of cooking skills in the associations of gender with social relationships and social capital. In the SEM analysis, social relationships and social capital were treated as latent variables estimated from neighborhood ties, frequency of meetings with friends, number of meetings with friends, frequent shared meals with friends, civic participation, and reciprocity (*n* = 15,207 because of missing values on the variables used to estimate the latent variables). Cooking skill, operationalized as the mean value of the seven cooking skill items, was treated as a continuous variable. Overall model fit was tested using the comparative fit index, the root mean square error of approximation, and the standardized root mean square residual. All analyses were conducted using Stata, Version 15 (Stata Statistical Software: Release 15. College Station, TX, USA: StataCorp LP).

## 3. Results

The participants’ characteristics are summarized in Appendix A. Women were about twice as likely as men to have a high level of neighborhood ties and to eat with their friends.

The associations between cooking skills and social relationships are shown in Table 1. The interaction effect between cooking skills and gender was significant: the relationships with all components of social relationships were higher among women than among men (*p* < 0.05 for the interaction). Women with a high level of cooking skills were 2.27 times (95% CI: 1.77–2.91) more likely to have a high level of neighborhood ties and 1.65 (95% CI: 1.20–2.27) times more likely to eat with friends, compared with women with middle/low-level cooking skills. High-level cooking skill was associated with a higher frequency and number of meetings with friends. Men with high-level cooking skills were 1.84 times (95% CI: 1.46–2.33) more likely to have a high level of neighborhood ties, compared with men with low-level cooking skills. For men, high-level cooking skill was associated with a higher frequency and number of meetings with friends. These associations remained significant after adjusting for prosocial behavior-related personality (Appendix A).

The associations between cooking skills and social capital are shown in Table 2. The interaction effect between cooking skills and gender was significant (*p* < 0.05 for the interaction). For women, high-level cooking skill was positively associated with all components of social capital, whereas the relationship between high-level cooking skill and social cohesion was non-significant for men. These associations remained significant after adjusting for prosocial behavior-related personality (Appendix A).

Compared with men, women had higher levels of social relationships and social capital except for social cohesion (Appendix A). Women were 3.01 times (95% CI: 2.76–3.29) more likely to have a high level of neighborhood ties and 2.47 times (95% CI: 2.20–2.78) more likely to eat with friends, compared with men. Women had a higher frequency and number of meetings with friends (coefficient = 0.34, 95% CI: 0.30–0.38 and coefficient = 0.67, 95% CI: 0.57–0.78), more civic participation (coefficient = 0.23, 95% CI: 0.19–0.27), and higher reciprocity (coefficient = 0.11, 95% CI: 0.10–0.13). However, women also had lower social cohesion compared with their male counterparts (coefficient = −0.04, 95% CI: −0.07 to −0.01).

Figure 1 shows the result of the SEM analysis for the association between gender and social capital including social relationships except for social cohesion. This SEM analysis demonstrated good model fit (likelihood-ratio test of the model, chi-square = 208.4, *p* < 0.001; comparative fit index = 0.991; root mean square error of approximation = 0.03; standardized root mean square residual = 0.016). The association between gender and social capital was partially mediated by cooking skill (from gender to cooking skill: standardized coefficient = 0.570, *p* < 0.001; from cooking skill to social relationships including social capital: standardized coefficient = 0.152, *p* < 0.001). The indirect effect was 26.2% of the total effect.

## 4. Discussion

To our knowledge, this is the first study to examine cooking skills as a modifiable determinant of social relationships and social capital. We found that, among older adults in Japan, a high level of cooking skill was positively associated with social relationships and social capital, and we identified significant interaction effects between cooking skill and gender on social relationships and social capital. We confirmed that women had higher levels of social relationships and social capital than men, and these associations were partially mediated by cooking skill.

Given that food plays a central role in connecting people in traditional Japanese culture [35], our results are plausible. Special meals for many rituals and celebrations throughout the year are handed down in various forms throughout Japan [36]. For events, people prepare special meals called *gyoujisyoku* and also hold “after parties” following the events [35]. Even outside of celebrations, many seasonal events connected with locally produced foods are held in communities [35]. For these events, people do not only eat together—they also make meals together, which strengthens friendships and cohesiveness [35]. Therefore, cooking skills are indispensable for these traditional and local events, and it is conceivable that people with higher levels of cooking skills will have more opportunities to play an important role in the community. We also found significant interaction effects between cooking skills and gender on social relationships and social capital: women are more likely to benefit from social relationships through a high level of cooking skills. This finding may be explained by women cooking more frequently than men [25], creating more opportunities for women to use their cooking skills.

In line with previous studies [10,15,18], we found that women were more likely than men to have strong social relationships and social capital. A nationally representative study in Ukraine showed that the gender difference in bonding social capital, which corresponds to the frequency/number of meetings with friends in our study, was explained by age and income [10]. Using SEM analysis, we found that cooking skill mediated 26% of the association between gender and social capital. Therefore, we have newly identified cooking skills as a factor contributing to explaining the gender differences in social capital.

Among the components of social capital, social cohesion was found to be weakly associated with cooking skills for women but not for men. In contrast to the other social relationships and social capital, social cohesion was the only variable that was lower in women than in men (Appendix A). Social cohesion, which is categorized as cognitive social capital rather than structural social capital, may have determinants that differ from those of other aspects of social relationships. A study conducted in the Netherlands showed that perceptions that one’s neighborhood is unsafe or unattractive and low SES were associated with low social cohesion but not with social networks (e.g., visiting neighbors, asking neighbors for advice) [37]. A study in the United States showed that neighborhood safety and SES were positively related to social cohesion [38]. Low-SES groups tend to be more pessimistic and express more feelings of unsafety and neighborhood problems compared with those with higher SES [37,39]. Therefore, neighborhood safety and SES may play key roles in cognitive social capital.

This study had several limitations. First, common method bias may have occurred because cooking skills, social relationships, and social capital were assessed via the self-report questionnaire. To address this common source bias, as a sensitivity analysis, we adjusted for mental health and prosocial behaviors, which are related to the tendency to respond to the questions. It would be useful to also collect information from a second person, such as a family member or experienced food preparer who could evaluate the participants’ cooking skills. Second, gender bias may have occurred because men tend to have higher self-esteem and more positive evaluations of their own abilities than women [40]. Men may overestimate their own cooking skills and women may underestimate theirs, in which case their relationship to social relationships and social capital may lead to underestimation. Third, there may be unmeasured confounding factors, such as regional characteristics. For example, in communities where cooking classes and events involving meal preparation are popular, residents will have more opportunities to build social relationships as well as improve their cooking skills. Regional characteristics may also influence participants’ evaluation of social capital. For example, participants may not feel as engaged in society as much as they should if they belong to an active community, and vice versa. In the future, indicators of community characteristics will need to be considered. Forth, because the JAGES survey study sites were not randomly selected, the generalizability of our findings to other populations in Japan is limited. Additionally, the cooking skill scale in this study is limited to Japanese culture. Therefore, the results of the study may be applicable only within Japan. In the future, cooking skill scales appropriate for each culture will need to be developed to evaluate aspects of health promotion in other countries. Finally, because this study was cross-sectional, causality could not be established: reverse causation is possible, and unmeasured factors may confound the examined associations. For example, having a low level of social relationships with neighbors/friends may reduce the chances of learning cooking skills, which may lead to poor cooking skills. However, more than half of the adult respondents learned most of their cooking skills from their mothers when they were teenagers [28].

## 5. Conclusions

Our study has produced novel findings regarding the associations of cooking skills with social relationships and social capital. Considering the health benefits of social relationships and social capital, our study is of great public health importance because it has demonstrated the importance of cooking skill, a factor that can be modified through intervention to improve social relationships and social capital.

## Figures and Tables

**Figure 1 ijerph-20-04633-f001:**
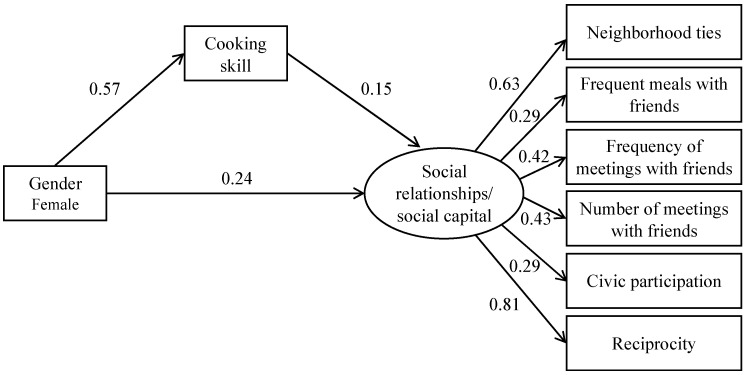
Path model of associations of gender with social relationships and social capital (*n* = 15,207). Path coefficients were standardized. All path coefficients were statistically significant (*p* < 0.05). Likelihood-ratio test of the model, chi-square = 208.4, *p* < 0.001; Comparative fit index = 0.991; Root Mean Square Error of Approximation = 0.03; Standardized Root Mean Square Residual = 0.016.

**Table 1 ijerph-20-04633-t001:** Associations of cooking skill with neighborhood ties, frequent meals with friends, and frequency and number of meetings with friends among older adults in Japan.

		Neighborhood Ties (Ref = Low-Level Ties)		Frequent Meals with Friends	Frequency of Meetings with Friends (*n*/week)	Number of Meetings with Friends (*n*/week)
		Middle-Level Ties	High-Level Ties			
		RRR (95% CI)	RRR (95% CI)	OR (95% CI)	Coefficient (95% CI)	Coefficient (95% CI)
Women						
Cooking skill	Middle/Low	ref	ref	ref	ref	ref
	High	**1.83 (1.53 to 2.20)**	**2.27 (1.77 to 2.91)**	**1.65 (1.20 to 2.27)**	**0.39 (0.28 to 0.50)**	**1.35 (1.08 to 1.63)**
Men						
Cooking skill	Low	ref	ref	ref	ref	ref
	Middle	**1.22 (1.06 to 1.41)**	**1.41 (1.10 to 1.80)**	1.06 (0.73 to 1.54)	**0.12 (0.03 to 0.22)**	**0.45 (0.20 to 0.70)**
	High	**1.34 (1.17 to 1.55)**	**1.84 (1.46 to 2.33)**	1.13 (0.79 to 1.62)	**0.22 (0.13 to 0.31)**	**0.60 (0.37 to 0.84)**

CI = confidence interval; OR = odds ratio; ref = reference group; RRR = relative risk ratio; SD = standard deviation. Boldface indicates statistical significance (*p* < 0.05). These models adjusted for age, education, annual normalized household income, marital status, and health status (cancer, heart disease, stroke, diabetes, hypertension, hyperlipidemia, and depressive symptoms).

**Table 2 ijerph-20-04633-t002:** Results of regression analyses of social capital according to the level of cooking skill among older adults in Japan.

		Civic Participation	Social Cohesion	Reciprocity
		0–5	0–3	0–3
		Coefficient (95% CI)	Coefficient (95% CI)	Coefficient (95% CI)
Women				
Cooking skill	Middle/Low	ref	ref	ref
	High	**0.24 (0.12 to 0.35)**	**0.11 (0.02 to 0.19)**	**0.28 (0.20 to 0.36)**
Men				
Cooking skill	Low	ref	ref	ref
	Middle	**0.13 (0.05 to 0.21)**	0.01 (−0.06 to 0.08)	**0.07 (0.01 to 0.14)**
	High	**0.18 (0.10 to 0.26)**	0.01 (−0.06 to 0.07)	**0.18 (0.12 to 0.24)**

CI = confidence interval; ref = reference group. Boldface indicates statistical significance (*p* < 0.05). These models adjusted for age, education, annual normalized household income, marital status, and health status (cancer, heart disease, stroke, diabetes, hypertension, hyperlipidemia, and depressive symptoms).

## Data Availability

The datasets generated and analysed during the current study are not publicly available due to ethical or legal restrictions but are available from the corresponding author upon reasonable request.

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
