# Peer review of "Associations of Cooking Skill with Social Relationships and Social Capital among Older Men and Women in Japan: Results from the JAGES"

_ijerph, 2023, doi:10.3390/ijerph20054633_

Round 1
Reviewer 1 Report
Thank you for allowing me to review the manuscript. The research deals with a valuable area of social capital among the elderly in Japan. However, the authors need to respond to the following points.
1. The study design is unclear.
Stating “Details of the study design have previously been described [25] (line 101)” at the beginning of the method section is a bad idea. The authors must explain the details of the study design to the readers as much as possible. The authors may refer to the Appendix or other literature after the detailed explanation, but not at the beginning of the section.
2. The variable definitions are difficult to follow.
The authors should create a table for the variable definitions. In the current form, the definitions are unclear and difficult to follow.
3. The possible sampling basis is unclear.
The authors should create a table for descriptive statistics.
4. The gender bias and/or peer effect should be more discussed.
As the methods are based on subjectiveness, the provided answers may suffer from gender bias and/or peer effect. Men may feel confident that they can do simple cooking, while women may not feel that way if women’s cooking skills are in general higher than those of men. In addition, responses to social relationships may be affected by residential characteristics. Subjects may not feel as engaged in society as much as they should if they belong to an active community, and vice versa. The authors should elaborate on these effects perhaps in the discussion section.
Reviewer 2 Report
Thank you for the very well written and detailed article on the influence of cooking skills on social relationships. A very interesting approach that definitely deserves more attention in health promotion from my point of view.
I have only a few comments that I would like you to consider when revising the manuscript.
From line 173: Covariates: The covariates that were included in the regression model are not discussed in the text below. In particular, the Geriatric Depression Scale. The reference to the questionnaire (Table 1) in line 174 is also incomprehensible to me. Was the table forgotten? Overall, it would be desirable to have a table in the supplement that shows the results of the regression analysis in detail, i.e. including the covariates.
Discussion and conclusion: The instrument used to assess cooking skills is very limited to Japanese/Asian cooking habits. It should be pointed out that the results of the analyses can only be related to Japan for the time being. Nevertheless, it would be desirable if this instrument could be adapted internationally in order to investigate this aspect of health promotion in other countries in the future.
Supplementary Materials (line 344): Here you list five tables. However, only four tables are included in the supplementary file. Please check and add the missing table.
Round 2
Reviewer 1 Report
Review Comments
Thank you for your responses. I think the quality of the manuscript has improved significantly. However, I do not understand why the authors include the subjects with a missing value. As in Table S1, a significant number of subjects did not provide their information on education, annual income, marital status, or depressive symptoms. It is very understandable for the subjects not to answer a certain question. In that case, however, the authors should exclude the subjects with a missing variable from the descriptive statistics table, because the estimation results that used the variable did not include the subjects anyway. Furthermore, this skews the descriptive statistics of all variables. Therefore, I think that the authors should not include subjects with a missing variable. If not, the authors should provide the justification.
